# Estimation of Stem-Solidness and Yield Components in Selected Spring Wheat Genotypes

**Mateusz Pluta [1], Danuta Kurasiak-Popowska [1] , Jerzy Nawracała [1], Jan Bocianowski [2] and Sylwia Mikołajczyk [1,*]**

1 Department of Plant Genetics and Breeding, Faculty of Agronomy, Horticulture and Bioengineering, Poznań University of Life Sciences, 60-632 Poznań, Poland; mateuszpluta01@gmail.com (M.P.); danuta.kurasiak-popowska@up.poznan.pl (D.K.-P.); jerzy.nawracala@up.poznan.pl (J.N.)

2 Department of Mathematical and Statistical Methods, Poznań University of Life Sciences, Wojska Polskiego 28, 60-637 Poznań, Poland; jan.bocianowski@up.poznan.pl

\* Correspondence: sylwia.mikolajczyk@up.poznan.pl

**Abstract:** Solid-stemmed wheat genotypes are better protected from damage caused by wheat stem sawfly (*Cephus pygmaeus* L.) larvae and at lower risk of lodging, as they are additionally strengthened. The aim of the study was to analyse the stem-solidness of fifty spring wheat cultivars with pith. A field experiment was conducted at the Agricultural Research Station Dłoń, Poland in the years 2012–2014. The method recommended by the International Union for the Protection of New Varieties of Plants (UPOV) and the methodology described by DePauw and Read were used to analyse the stem-solidness. The statistical analysis of the results showed that the stems of the wheat cultivars differed in their, therefore, they were divided into seven classes. There were nine Polish cultivars, two genotypes from Canada (BW 597 and AC Elsa) and one Portuguese genotype (I 836) with hollow stems. There were only nine solid-stemmed cultivars. Both methodologies were used to assess the filling of the stem in the whole plant upon analysis of its filling at the cross-section of the first internode. Both methods gave the same results. The DePauw and Read methodology showed that the internodes in the lower part of the plants were filled to the greatest extent. The same genotypes collected in the consecutive years of the study differed in the filling of their stems with pith. These differences were influenced by the environmental conditions.

**Keywords:** spring wheat; solid stem; hollow stem; *Cephus cinctus*

## 1. Introduction

Most wheat cultivars grown in Europe have a hollow stem, and only a few varieties are solid stemmed. Depending on the genotype, the expression of this trait ranges from 0 to 100%. The inside of the stem is filled with the main parenchyma, which forms the pith [1]. This structure resembles a honeycomb and consists of thin-walled cells of equal size that are loosely adjacent to each other [2].

Stems filled with pith are better protected from the damage caused by wheat stem sawfly (*Cephus pygmaeus* L.) larvae and at a lower risk of lodging, as they are additionally strengthened. The wheat stem sawfly is a pest of cereals and grasses, which is commonly found in the temperate climate zone. Between late May and early June, i.e., at the wheat flowering time, adult wheat stem sawflies emerge [3]. During this period, females lay eggs inside the uppermost internode of wheat [4–6]. The larvae that hatch from these eggs eat the tissues inside the stem, thus deteriorating the seed quality. Finally, they cut the stems and cause wheat lodging [7,8]. The survival rate of juvenile wheat stem sawflies is higher in monocultures and no-till farming.

Most studies on wheat stem filling are conducted in Canada and the United States, because the local wheat stem sawfly species (*Cephus cinctus* Northon) causes significant

damage to crops. In 2009, the inspection of wheat producers revealed that wheat stem sawflies caused a yield loss of 10–25% in North Dakota, USA [9].

The wheat stem sawfly as a pest is not as important in Poland as in southern Europe [10]. According to the Plant Protection Institute—National Research Institute, in recent years, the wheat stem sawfly population has increased in Poland due to global warming, which results in mild and warm winters, during which larvae are more likely to survive. The survival rate of juvenile wheat stem sawflies is even higher in widespread, long-term monocultures and no-till farming, which is increasingly popular. Between 1966 and 1995, the average damage to wheat plantations caused by wheat stem sawflies in Poland amounted to 0.3–3%. During the period under analysis, the greatest damage in individual voivodeships ranged up to 8.5%, whereas in some areas it was even up to 21% [11].

Stems filled with pith not only increase the resistance to *Cephus pygmaeus* L. but also additionally stiffens the plant, protecting it against the lodging caused by environmental factors. Cultivars with an increased resistance to lodging [12] and wheat stem sawflies could be obtained by incorporating the pith-filled stem trait into breeding materials. As the stem-solidness is heavily modified by environmental conditions, it is difficult to use the genotypes that are unstable with this trait in crossbreeding programmes. Solid-stemmed wheat cultivars exhibit trait expression differences in individual years of cultivation due to the interaction of genes and environmental factors [13]. The amount of pith in the stem is mainly influenced by insolation, temperature, humidity, the content of soil nutrients [14], and plant density [15,16]. Beres et al. [16] found the highest amount of pith in the Lillian cultivar at the lowest plant density per $m^2$. The authors also proved that this cultivar had a high and stable pith filling when the sowing density ranged from 205 to 350 kernels per $m^2$. Beres et al. [17] published more accurate results, where the highest filling of the stem with the pith was observed at a density of 100 plants per $m^2$. They found that stem-solidness increased along with the nitrogen fertiliser dose up to 60 kg N ha$^{-1}$, but it decreased at higher nitrogen doses. The results of the study by Nilsen et al. [18] confirmed the fact that the degree stem-solidness in all solid-stemmed cultivars tended to decrease as the sowing density increased.

Although the cultivation of solid-stemmed cultivars provides the best protection from wheat stem sawflies, for a long time, researchers believed that this trait negatively affected the yield of seeds [19]. Actually, the research does not confirm that the presence of steam-solidness has a negative effect on the wheat yield [20–22].

The aim of the study was the analysis of the stem-solidness of fifty spring wheat cultivars grown in Polish environmental conditions.

## 2. Materials and Methods

### 2.1. Plant Material

The study was conducted on fifty spring wheat cultivars and lines, which differed in stem-solidness. This collected genotypes included all major solid-stemmed spring wheat genotypes. The oldest cultivar was from 1918 and the newest from 2011. The collection included genotypes from the following countries and regions of the world: Asia Minor, the Czech Republic, Greece, the Netherlands, Canada, Germany, New Zealand, Poland, Portugal, Russia (the Soviet Union), Syria, Uruguay, and the USA (Table S1). The seeds of these genotypes were obtained from gene banks as well as from Polish and foreign breeders.

### 2.2. Field Experiment

A field experiment was conducted at the Agricultural Research Station Dłoń, Poznań University of Life Sciences, Poland (51°41′22″ N 17°04′23″ E) from 2012 to 2014. Soybean was used as a forecrop in the experiments. In autumn, the soil was ploughed for winter; in spring, it was smoothed, tilled with a cultivator, and treated with the following mineral fertilisers: Lubofoska (5% N, 10% P, 15% K)—200 kg ha$^{-1}$, and ammonium nitrate (34% N)—100 kg ha$^{-1}$. Subsequent doses of the nitrogen fertiliser (ammonium nitrate 34% N)—150 kg ha$^{-1}$—were applied at the phases of stem elongation and heading.

The experiment was conducted in a randomised complete block design in three replicates—in total there were 150 plots with an area of 1 m² each. There were 350 plants in each plot. The plants were sown with a plot seeder at a spacing of 20 cm. The experiments were started established on 11 April 2012, 23 April 2013 and 4 April 2014.

Each year in August, 30 plants at the phase of full maturity were harvested from each plot for biometric assessment. The stems of the plants were assessed for filling with pith in the uppermost internode and in the entire plant. Apart from that, the following morphological traits and elements of the following yields components were assessed: the plant height, the number of stems, the number of spikes, the spike length, the number of spikelets per spike, the number of seeds per spike, the number of seeds per plant, the weight of seeds per spike, the weight of seeds per plant, and the thousand kernel weight.

### 2.3. Assessment of Stem-Solidness

The main stem of 30 plants selected at random from each plot was visually assessed for its solidness. The following two methods were used for the assessment:

The method recommended by the International Union for the Protection of New Varieties of Plants (UPOV)—the uppermost internode was cut at half-length and its filling was rated according to the following degrees (Figure 1).

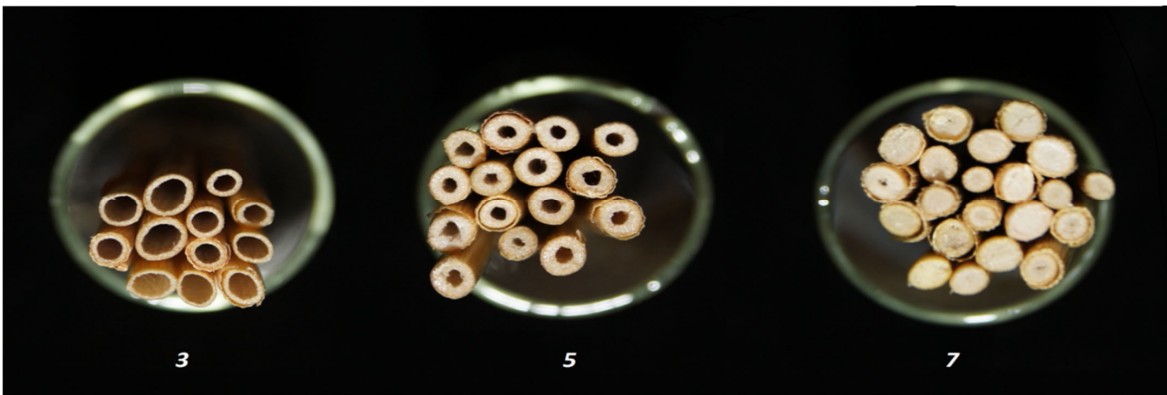

**Figure 1.** The assessment of the stem-solidness according to the methodology recommended by the UPOV: 3—thin pith = hollow, 5—intermediate pith, 7—thick pith = solid (Photo by M. Pluta).

The method described by DePauw and Read [23]—five consecutive internodes were cut at half-length and the cross-section of the stem was rated 1–5. The rating of the filling of the stem of the whole plant was the total of the ratings of the five internodes. A plant with a hollow stem scored 5 points, whereas a solid stem scored 25 points (Figure 2).

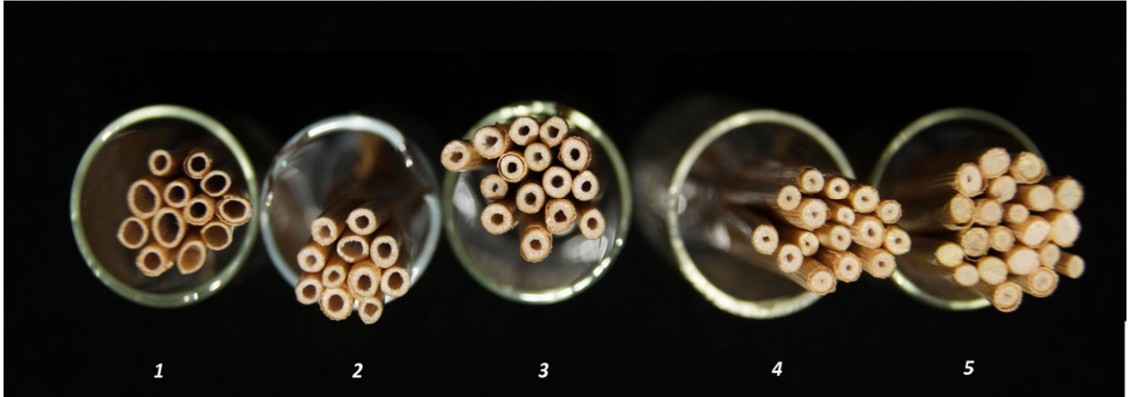

**Figure 2.** The assessment of the stem-solidness according to the methodology developed by DePauw and Read (1982): 1—hollow pith (0% filled), 2—25% filled, 3—50% filled, 4—75% filled, 5—solid stem (100% filled) (Photo by M. Pluta).

*2.4. Statistical Analysis*

The normality of the distribution of the ten quantitative traits (plant height, number of stems, average number of spikes, average spike length, average number of spikelets per spike, average number of seeds per spike, average number of seeds per plant, seed weight per plant, thousand kernel weight, average stem filling with pith) and the multivariate normal distribution were tested. A two-way (year, genotype) multivariate analysis of variance (MANOVA) was made. Following this, Combined Analysis of Variance were performed in order to verify the null hypotheses of a lack of year and genotype effects as well as year × genotype interaction effect in terms of the values of the ten observed traits, independently for each trait. The arithmetic means and standard deviations were calculated. Moreover, Fisher's least significant differences (LSDs) were estimated at a significance level of α = 0.05. Additionally, the variability of values of the traits was presented in boxplots. The relationships between the observed traits were estimated using Pearson's correlation coefficients for each year independently.

A multi-level regression analysis was performed to estimate whether different weather conditions in the following years were important for filling the stem with the core. This analysis was chosen due to the nested nature of the data. The maximum likelihood (ML) estimation used in multi-level regression analysis is better at dealing with the nested nature of the data and the unbalanced system.

In the first step, the so-called empty model, i.e., taking into account only information about data nesting. In the second step, two instrumental variables representing subsequent years of experience were added to the model. The first year of experiment (2012) was selected as the comparative year. Using the data collected from the assessment of the degree of stem filling with the core according to the methodology used in the USA, from individual internodes of 50 spring wheat genotypes from three years of observation, an attempt was made to group and assign plants to clusters depending on the degree of core filling of the stem.

The variables used for these calculations were the results of the evaluation of the stem core filling 1 (main) and the average of the filling of the stalks from 2 to 15, totalled for each plant. This treatment was performed because individual plants had a different number of blades. The data prepared in this way were subjected to grouping of plants by agglomeration cluster analysis using the Ward's method and using the Caliński–Harabasz Index. Then, using the same method, varieties were grouped into clusters depending on from the degree of core filling of the stem.

The GenStat 18 package and the Stata 13 program were used for statistical calculations.

## 3. Results

*3.1. Weather Conditions*

The average monthly temperatures in all the growing seasons were higher than between 1961 and 2011 (Figure 3). The average monthly temperatures from 1961–2011 during the growing season were 8.5 °C in April, 13.8 °C in May, 17.0 °C in June, 18.7 °C in July, and 18.2 °C in August. During the wheat growing season (April–August) the biggest difference in monthly temperatures was noted in May 2012 (4 °C), whereas the smallest was in July (1.3 °C). During the vegetation period in 2013 and 2014, the average monthly temperature in July was, respectively, 4.6 and 5.2 °C higher than the average temperatures between 1961 and 2011.

The average monthly rainfall in the years 1961–2011 during the growing season was 31.3 mm in April, 56.8 mm in May, 64.3 mm in June, 81.1 mm in July, and 67.5 mm in August. In 2012, there was lower monthly rainfall. In the spring of 2013, there was lower monthly rainfall than the average total rainfall between 1961 and 2011. In 2013, the experimental plots were flooded for almost one month due to the high rainfall in late May (97.5 mm) and early June (85.2 mm). In 2014, at the initial stage of wheat growth, there was a higher rainfall than the average total rainfalls between 1961 and 2011, whereas in the summer there was a drought.

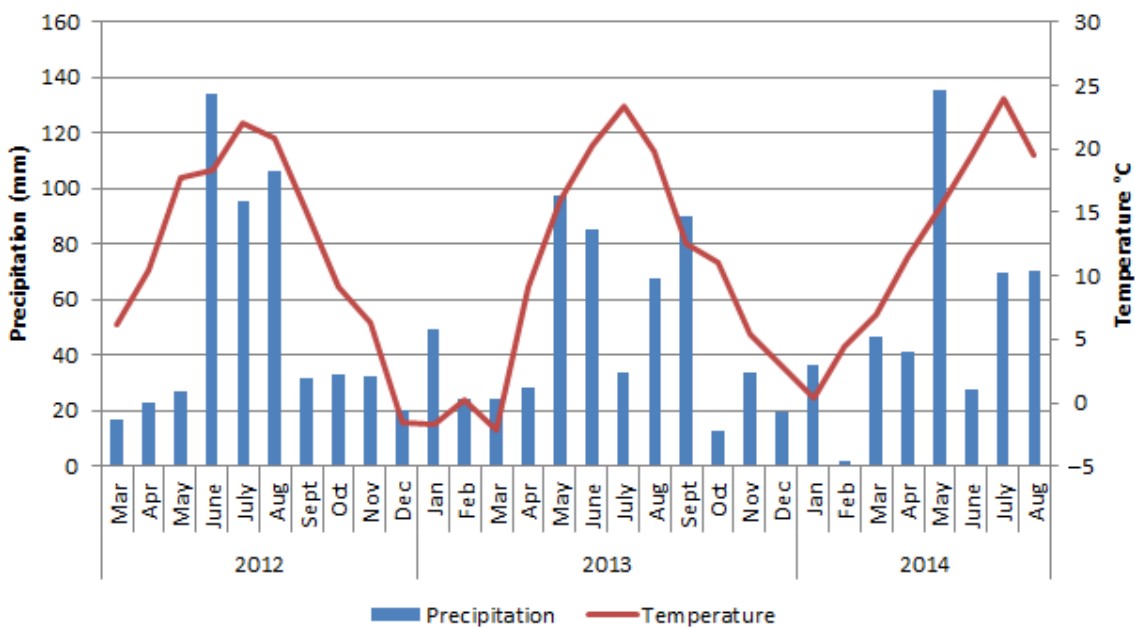

**Figure 3.** The distribution of rainfall and mean temperatures during the field experiment in Dłoń, Poland.

### 3.2. Analysis of Degree of Stem-Solidness according to UPOV Methodology

All the observed traits had a normal distribution. The results of the MANOVA performed indicated that all the years (Wilk's $\lambda$ = 0.0014987; $F$ = 223.48; $p$ < 0.0001), cultivars (Wilk's $\lambda$ = 0.000234; $F$ = 250.92; $p$ < 0.0001) and year × cultivar interaction (Wilk's $\lambda$ = 0.0000176; $F$ = 309.07; $p \leq$ 0.0001) were significantly different with regard to all of the observed traits. A combined ANOVA indicated that the main effects of year and genotype as well as year × genotype interaction were significant for all 18 observed traits (Table S2).

As much as 50% of the 50 spring wheat cultivars analysed for the filling of the uppermost internode with pith were classified as hollow-stemmed. This group included all Polish cultivars (hollow-stemmed by definition) and 14 genotypes that originated from other countries. Thirteen cultivars were classified into the group with an intermediate pith expression in the uppermost internode. A total of 12 genotypes were classified as solid-stemmed, because their uppermost internode was significantly filled with pith (Table 1).

The highest average filling of the uppermost internode with pith was observed in the growing season of 2013 (4.7 points) (Table 1). In that year, 18 genotypes were classified as hollow-stemmed, 20 were classified into the group with intermediate pith filling, and 12 into the solid-stemmed group. In the growing season of 2014, the uppermost internode of the plants was filled with pith to the lowest extent (3.79). As many as 35 wheat cultivars were classified into the hollow-stemmed group, 12 were classified into the group with intermediate pith filling, and the uppermost internode of only three cultivars was rated as filled with pith (CI 7033, Lew, Ruzyska II). Different reactions of the cultivars to individual years of the experiment were confirmed by a significant cultivar × year interaction.

The analysis of the average values of the degree of filling of the uppermost internode with pith in individual cultivars during the three-year period of the research in Poland showed that 22 cultivars could be classified as hollow-stemmed, seven cultivars were filled with pith to a large extent, whereas the other 21 genotypes could be classified as intermediately filled with pith in the uppermost internode. This trait was unstable in the 3353 and Alentejano cultivars, as their uppermost internodes were differently filled with pith in the individual years of the study.

**Table 1.** The average filling of the uppermost internode of the stem with pith in 50 spring wheat genotypes (Agricultural Research Station Dłoń, 2012–2014) according to the methodology recommended by the UPOV.

| No. | Genotype | Pith in Uppermost Internode | | | |
|---|---|---|---|---|---|
| | | 2012 | 2013 | 2014 | Mean Score of Genotype |
| 1. | Arabeska | 3.00 | 3.00 | 3.00 | 3.00 |
| 2. | Bombona | 3.00 | 3.00 | 3.00 | 3.00 |
| 3. | Kandela | 3.01 | 3.13 | 3.00 | 3.05 |
| 4. | Katoda | 3.01 | 3.01 | 3.00 | 3.01 |
| 5. | Łagwa | 3.00 | 3.00 | 3.00 | 3.00 |
| 6. | Nawra | 3.00 | 3.00 | 3.00 | 3.00 |
| 7. | Ostka Smolicka | 3.00 | 3.00 | 3.00 | 3.00 |
| 8. | Parabola | 3.00 | 3.00 | 3.00 | 3.00 |
| 9. | Radocha | 3.00 | 3.00 | 3.00 | 3.00 |
| 10. | Raweta | 3.01 | 3.00 | 3.00 | 3.00 |
| 11. | Waluta | 3.00 | 3.00 | 3.00 | 3.00 |
| 12. | 431 | 3.75 | 5.27 | 3.44 | 4.15 |
| 13. | Ruzynska II | 4.70 | 5.96 | 6.81 | 5.82 |
| 14. | 404 | 3.21 | 3.26 | 3.00 | 3.16 |
| 15. | Tybalt | 5.77 | 6.71 | 3.80 | 5.42 |
| 16. | BW 597 | 3.01 | 3.00 | 3.00 | 3.00 |
| 17. | Chester | 5.32 | 4.86 | 3.23 | 4.47 |
| 18. | Chinook | 3.85 | 4.98 | 3.98 | 4.27 |
| 19. | Cypress | 4.69 | 5.61 | 3.39 | 4.56 |
| 20. | Manitou | 3.00 | 3.00 | 3.00 | 3.00 |
| 21. | Marquis | 3.37 | 5.20 | 3.31 | 3.96 |
| 22. | AC Abbey | 5.85 | 6.31 | 5.86 | 6.00 |
| 23. | AC Eatonia | 4.28 | 4.72 | 4.66 | 4.55 |
| 24. | AC Elsa | 3.00 | 3.00 | 3.00 | 3.00 |
| 25. | Canuck | 6.95 | 6.25 | 3.68 | 5.63 |
| 26. | Lancer | 5.67 | 6.09 | 3.05 | 4.94 |
| 27. | Leader | 3.84 | 5.09 | 3.18 | 4.04 |
| 28. | Lillian | 4.99 | 5.20 | 3.21 | 4.47 |
| 29. | Rescue | 6.49 | 6.12 | 5.87 | 6.16 |
| 30. | Carola | 6.35 | 6.95 | 5.88 | 6.39 |
| 31. | Heines Germania | 3.01 | 3.00 | 3.00 | 3.00 |
| 32. | Solid Straw T. Varia | 4.14 | 4.04 | 3.00 | 3.73 |
| 33. | 3353 | 6.60 | 4.21 | 3.08 | 4.63 |
| 34. | Alentejano | 3.81 | 6.66 | 4.00 | 4.82 |
| 35. | Beirao | 6.09 | 4.23 | 3.00 | 4.44 |
| 36. | Cl 7033 | 5.77 | 6.95 | 6.53 | 6.42 |
| 37. | Cltr 7027 | 7.00 | 6.15 | 5.84 | 6.33 |
| 38. | Cltr 7028 | 6.98 | 5.29 | 4.94 | 5.74 |
| 39. | H N ROD 5 13750 | 3.25 | 3.28 | 3.06 | 3.20 |
| 40. | I 836 | 3.00 | 3.00 | 3.00 | 3.00 |
| 41. | Leda Collection A47 | 6.47 | 5.84 | 4.46 | 5.59 |
| 42. | 401 | 6.67 | 5.87 | 5.47 | 6.00 |
| 43. | Americano 44D | 3.04 | 5.12 | 3.00 | 3.72 |
| 44. | Fortuna | 6.53 | 5.81 | 4.64 | 5.66 |
| 45. | Tioga | 5.00 | 6.19 | 3.25 | 4.81 |
| 46. | Glenman | 5.70 | 4.58 | 4.08 | 4.79 |
| 47. | Lew | 6.87 | 5.99 | 6.04 | 6.30 |
| 48. | MT 776 | 6.46 | 6.35 | 4.56 | 5.79 |
| 49. | MTMSSF-88 | 3.71 | 5.13 | 3.04 | 3.96 |
| 50. | Sawtana | 5.39 | 6.46 | 3.15 | 5.00 |
| mean | | 4.53 | 4.70 | 3.79 | |

$LSD_{0.05}$ for cultivars = 0.122, $LSD_{0.05}$ for years = 0.035, $LSD_{0.05}$ for cultivar × year interaction = 0.280.

### 3.3. Analysis of Stem-Solidness according to Methodology Developed by DePauw and Read (1982)

The degree of the stem-solidness in the whole plant was also assessed with the methodology developed by DePauw and Read [23] in the USA (Table 2). The analysis of the mean filling of the first (uppermost) internode with pith according to the five-point American scale showed that the following five genotypes produced the most parenchyma: AC Abbey, Carola, CI 7033, CItr 7027, and Lew. By contrast, all Polish wheat cultivars as well as the following varieties were hollow stemmed in the first internode: 404, BW 597, AC Elsa, Heines Germania, H N ROD 5 13750, I 836, and Manitou. The most parenchyma in the first internode was observed in 2013 (2.64).

**Table 2.** The classification of cultivars according to the stem-solidness based on the methodology developed by DePauw and Read (1982).

| No. | Genotype | Number of Plants | Degree of Stem Filling with Pith | | | | Class |
|---|---|---|---|---|---|---|---|
| | | | Hollow (%) | Intermediate (%) | Solid (%) | Total Score | |
| 1 | Waluta | 247 | 100 | 0 | 0 | 5 | 1 |
| 2 | I 836 | 257 | 93 | 7 | 0 | 5 | 1 |
| 3 | Katoda | 246 | 98.78 | 1.22 | 0 | 5.1 | 1 |
| 4 | Bombona | 230 | 99.13 | 0.87 | 0 | 5.7 | 1 |
| 5 | Parabola | 217 | 89.86 | 10.14 | 0 | 5.8 | 1 |
| 6 | AC Elsa | 183 | 87.98 | 12.02 | 0 | 5.8 | 1 |
| 7 | Ostka Smolicka | 232 | 83.19 | 16.81 | 0 | 5.9 | 1 |
| 8 | Radocha | 227 | 100 | 0 | 0 | 5.9 | 1 |
| 9 | BW 597 | 197 | 89.85 | 10.15 | 0 | 5.9 | 1 |
| 10 | Kandela | 228 | 95.18 | 4.83 | 0 | 6.1 | 1 |
| 11 | Raweta | 238 | 81.51 | 18.49 | 0 | 6.4 | 1 |
| 12 | Łagwa | 226 | 96.02 | 3.98 | 0 | 7 | 1 |
| 13 | Heines Germania | 199 | 62.31 | 37.19 | 0.50 | 6.2 | 2 |
| 14 | Manitou | 208 | 71.64 | 28.37 | 0 | 6.3 | 2 |
| 15 | Nawra | 204 | 75 | 24.51 | 0.49 | 7.6 | 2 |
| 16 | Americano 44D | 246 | 47.56 | 52.03 | 0.41 | 5.8 | 3 |
| 17 | MTMSSF-88 | 231 | 46.75 | 38.96 | 14.29 | 6.2 | 3 |
| 18 | Arabeska | 249 | 52.21 | 47.79 | 0 | 6.5 | 3 |
| 19 | Solid Straw T. Varia | 229 | 39.74 | 55.46 | 4.80 | 7.3 | 3 |
| 20 | H N ROD 5 13750 | 243 | 38.27 | 51.85 | 9.88 | 7.5 | 3 |
| 21 | 404 | 206 | 37.38 | 60.19 | 2.43 | 9.5 | 3 |
| 22 | Lillian | 180 | 32.22 | 37.22 | 30.56 | 7.2 | 4 |
| 23 | Beirao | 240 | 23.75 | 49.17 | 27.08 | 7.4 | 4 |
| 24 | Marquis | 206 | 22.33 | 54.85 | 22.82 | 13.1 | 4 |
| 25 | Glenman | 245 | 3.27 | 50.61 | 46.12 | 11.5 | 5 |
| 26 | 431 | 230 | 16.09 | 43.04 | 40.87 | 14.3 | 5 |
| 27 | Ruzynska II | 214 | 0.47 | 63.55 | 35.98 | 14.6 | 5 |
| 28 | Cl 7033 | 228 | 5.26 | 61.4 | 33.33 | 14.7 | 5 |
| 29 | 401 | 214 | 2.34 | 47.66 | 50.00 | 16.5 | 5 |
| 30 | Lew | 245 | 0 | 59.18 | 40.82 | 17.9 | 5 |
| 31 | Sawtana | 226 | 23.89 | 13.27 | 62.83 | 7.6 | 6 |
| 32 | Lancer | 214 | 24.77 | 13.55 | 61.68 | 7.8 | 6 |
| 33 | 3353 | 222 | 23.87 | 18.92 | 57.21 | 7.9 | 6 |
| 34 | Tioga | 250 | 20 | 18 | 62.00 | 8 | 6 |
| 35 | Leader | 225 | 21.33 | 29.78 | 48.89 | 8.3 | 6 |
| 36 | Alentejano | 253 | 13.04 | 31.23 | 55.73 | 10.8 | 6 |
| 37 | Fortuna | 234 | 7.69 | 26.5 | 65.81 | 12.2 | 6 |
| 38 | Chester | 239 | 21.34 | 23.85 | 54.81 | 13.4 | 6 |
| 39 | Leda Collection A47 | 221 | 9.05 | 28.96 | 61.99 | 13.4 | 6 |
| 40 | Chinook | 210 | 11.43 | 36.19 | 52.38 | 14.5 | 6 |
| 41 | Cypress | 195 | 14.36 | 19.49 | 66.15 | 16.1 | 6 |
| 42 | MT 776 | 221 | 7.69 | 20.36 | 71.95 | 12.4 | 7 |
| 43 | AC Abbey | 228 | 2.19 | 24.56 | 73.25 | 14.8 | 7 |
| 44 | Tybalt | 221 | 2.26 | 19.91 | 77.83 | 16.7 | 7 |
| 45 | AC Eatonia | 223 | 0 | 28.7 | 71.30 | 16.7 | 7 |
| 46 | Carola | 250 | 0 | 12.8 | 87.20 | 16.7 | 7 |
| 47 | Cltr 7028 | 210 | 0 | 18.57 | 81.43 | 17.5 | 7 |
| 48 | Rescue | 255 | 2.35 | 23.53 | 74.12 | 17.9 | 7 |
| 49 | Canuck | 258 | 10.85 | 12.4 | 76.74 | 18.4 | 7 |
| 50 | Cltr 7027 | 223 | 0 | 5.83 | 94.17 | 20.8 | 7 |

The highest average filling of the stem with pith in the four consecutive internodes in all the genotypes was observed in 2012. The filling of the stem with pith tended to increase toward the bottom of the plant. The lower the internodes were located, the more filled they were and the higher they scored on the five-point scale.

The measurements of the degree of the filling of the other internodes with pith showed that the second internode in 21 spring wheat cultivars had practically no parenchyma tissue inside. The following cultivars had no pith in the third internode: Americano 44D, AC Elsa, BW 597, Heines Germania, I 836, Manitou, and nine Polish cultivars. The analysis of the average filling of the fourth internode with pith showed that only three genotypes had a high degree of filling (AC Eatonia, CItr 7027, and Tybalt), whereas four wheat cultivars (Bombona, I 836, Katoda, and Waluta) were hollow inside. The morphological analyses of the spring wheat cultivars showed that, in 2012, four genotypes did not develop the fifth internode (Beirao, Nawra, Radocha, Ruzynska II). The analysis of the results showed that on average, during the three years of the research, AC Eatonia (4.65 points) and CItr 7027 (4.54 points) were the cultivars whose fifth internode had the most parenchyma tissue. Bombona and Waluta were the cultivars that did not develop pith in this internode. The other genotypes developed intermediate structures.

The analysis of variance showed significant differences between the cultivars and years as well as cultivar × year interactions within individual internodes as well as the average total degree of stem in the whole plant (Table S2). Figure 4 shows the range of variability in the stem solidness between 2012 and 2014. The cultivars with intermediate stem filling exhibited the greatest range of variability.

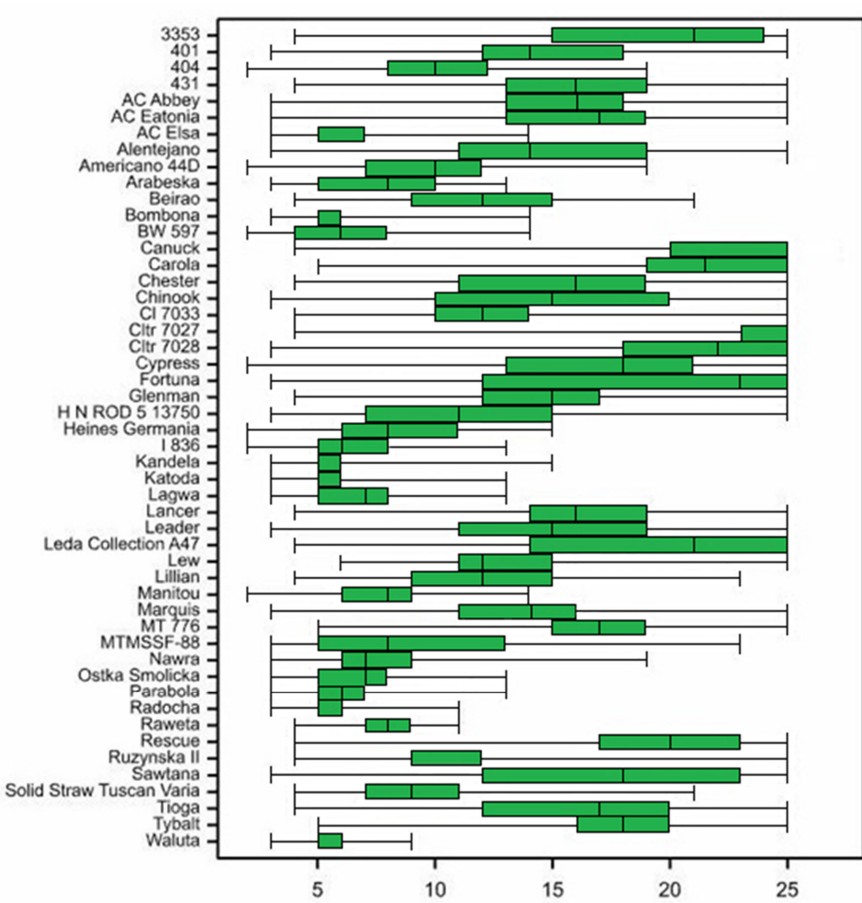

**Figure 4.** The range of variability of the stem-solidness in 50 spring wheat genotypes (Agricultural Research Station Dłoń, 2012–2014) assessed according to the methodology developed by DePauw and Read.

### 3.3.1. Classification of Cultivars according to Stem-Solidness

The agglomerative clustering analysis with the Ward method showed that the division of the cultivars into seven classes would be the most optimal method of grouping them according to the degree of their stem solidness (Table 3).

**Table 3.** The selection of the optimal number of classes to group 50 spring wheat genotypes with the Ward method.

| Number of Classes | Caliński–Harabasz Index (CHk) |
|---|---|
| 2 | 110.39 |
| 3 | 104.91 |
| 4 | 153.66 |
| 5 | 156.79 |
| 6 | 169.74 |
| 7 | 180.73 |
| 8 | 177.36 |

Seven classes of clusters were distinguished among the cultivars according to the degree of their stem solidness. The percentage of the plants allocated to individual clusters of filling (hollow, intermediate, and solid) was specified (Table 4). The plants were divided into hollow-, intermediate-, and solid-stemmed ones on the basis of all the plants analysed in the experiment.

**Table 4.** Descriptive statistics for genotype classes of 50 spring wheat genotypes grouped with the Ward method.

| | Cultivar Class | N [#] | Plant Class | % of Plants in Class | Variance | Standard Deviation |
|---|---|---|---|---|---|---|
| 1 | Hollow-stemmed genotypes | 12 (24%) | Hollow | 92.87 | 41.44 | 6.44 |
| | | | Intermediate | 7.13 | 41.44 | 6.44 |
| | | | Solid | 0 | 0 | 0 |
| 2 | Nearly hollow-stemmed genotypes | 3 (6%) | Hollow | 69.65 | 43.20 | 6.57 |
| | | | Intermediate | 30.02 | 42.23 | 6.50 |
| | | | Solid | 0.33 | 0.08 | 0.29 |
| 3 | Genotypes of less than intermediate filling level | 6 (12%) | Hollow | 43.65 | 36.35 | 6.03 |
| | | | Intermediate | 51.05 | 52.29 | 7.23 |
| | | | Solid | 5.30 | 32.45 | 5.70 |
| 4 | Genotypes of intermediate filling level | 3 (6%) | Hollow | 26.10 | 28.61 | 5.35 |
| | | | Intermediate | 47.08 | 80.99 | 9.00 |
| | | | Solid | 26.82 | 15.03 | 3.88 |
| 5 | Genotypes of greater than intermediate filling level | 6 (12%) | Hollow | 4.57 | 35.51 | 5.96 |
| | | | Intermediate | 54.24 | 68.85 | 8.30 |
| | | | Solid | 41.19 | 38.21 | 6.18 |
| 6 | Nearly solid-stemmed genotypes | 11 (22%) | Hollow | 17.34 | 40.39 | 6.36 |
| | | | Intermediate | 23.61 | 56.98 | 7.55 |
| | | | Solid | 59.04 | 31.49 | 5.61 |
| 7 | Solid-stemmed genotypes | 9 (18%) | Hollow | 2.82 | 15.12 | 3.89 |
| | | | Intermediate | 18.52 | 50.21 | 7.09 |
| | | | Solid | 78.66 | 59.34 | 7.70 |

[#] N—Number of genotypes allocated to individual clusters.

The grouping of the cultivars according to the filling of their stems with pith confirmed the actual state of the expression of this trait in individual cultivars. Table 2 shows the allocation of individual cultivars to specific clusters. The information in the table shows very precisely the expression of the stem-filled-with-pith trait in individual genotypes on the Polish environmental conditions. The first class of hollow-stemmed genotypes

included nine Polish cultivars, two genotypes from Canada (BW 597 and AC Elsa), and one Portuguese genotype (I 836). The total score of these genotypes for the filling of their stems with pith ranged from five to seven. In classes two and three, the filling of the stem with pith ranged from 5.8 to 9.5 points. In classes 4–6, the score of the genotypes ranged from 7.2 to 17.9 points. The solid-stemmed class included four Canadian genotypes (AC Abbey, AC Estonia, Canuck, and Rescue), two Portuguese genotypes (Cltr 7027 and Cltr 7028), one American genotype (MT 776), one Dutch genotype (Ttybalt), and one German genotype (Carola). The degree of stem-solidness in this group ranged from 12.4 (MT 776) to 20.8 points (Cltr 7027).

### 3.3.2. Differences in Stem-Solidness by Year

The calculations showed that differences in the filling of individual stems with pith were largely cultivar-dependent (about 57% of the differences between the stems was cultivar-specific) (Table S3). Plants of the same cultivar also differed from each other in the filling of their stems with pith, but these differences explained much less variation in the filling.

The analyses showed (the model with years) that the filling of the stem with pith depended on the weather conditions, which differed in the subsequent years of the experiment (Table S4). In 2013 and 2014, the stem-solidness was lower than in 2012. It should be recalled that April and May 2012 were dry with relatively high temperatures. In 2013, the average filling of the stems with pith was about 0.2 point lower than in 2012, whereas in 2014, it was about one point lower than in 2012. Only five cultivars (401, AC Eatonia, Bombona, Łagwa, and Rescue) showed the stability of stem-solidness, regardless of the weather conditions during the wheat growth.

### 3.4. Yield Components Traits

The results of biometric analyses can be found in the supplement to this study (Tables S5–S7). The genotypes, years, and genotypes × year interaction were statistically significant for all the observed traits.

In 2012, the height of fifteen genotypes exceeded 100 cm. The plants of the following cultivars were the tallest: Americano 44D, CItr 7027, and I 836 (Table S5). In 2013, the weather conditions did not favour the development of spring wheat. In consequence, the average height of all the 50 genotypes analysed in the study was the lowest, i.e., only 68.6 cm. In 2014, the plants of 31 spring wheat genotypes exceeded the average height of 100 cm. BW 597 was the shortest genotype in 2013 and 2014. Its average height was also the lowest during the three years of the research.

Table S5 shows the average number of stems in the 50 spring wheat cultivars. In 2012, there were 5.3 stems per plant on average. This trait ranged from 3.5 to 7.7. In 2013, the weather conditions were not conducive to good branching of spring wheat plants; therefore, the number of stems per plant ranged from 1.07 to 2.95 in individual cultivars. In the growing season of 2014, the average number of stems per plant was 1.67.

During the three-year study, the following cultivars were characterised by the greatest average spike length: Carola, Glenman, Ostka Smolicka, and Raweta. The BW 597 cultivar produced the smallest spikes (Table S6). The Ostka Smolicka cultivar had the most seeds per spike, whereas the BW 597 cultivar had the fewest. The largest average number of kernels per spike was found in the Katoda, Parabola, and Radocha cultivars. The BW 597 cultivar yielded the fewest kernels per spike. The Polish spring wheat cultivars set the most seeds per plant, i.e., over 70. The Parabola cultivar was characterised by the largest and the most stable seed weight per spike during the three years of the study. On the other hand, the BW 597 cultivar produced seeds of the lowest weight in all the three years of the study. The Ostka Smolicka cultivar was characterised by the highest mean weight of seeds per plant during the three years of the study (Table S7). The comparison of the cultivars in terms of the thousand kernel weight (TKW) showed that in 16 of them, the average TKW

was greater than 40 g during the three years of the study. The Canadian cultivar BW 597 was characterised by the lowest TKW, i.e., lower than 30 g.

### 3.5. Relationships between Observed Traits

The results of correlation analysis between traits observed in 2012, 2013, and 2014 are presented in the Tables S8–S10. In all three years of study, positive correlation coefficients were observed for the pairs: number of stems and average number of spikes, average spike length and average number of spikelets per spike, pith in uppermost internode and average stem filling with pith in total internodes, average spike length and average number of seeds per spike, number of stems and average number of seeds per plant, number of stems and seed weight per plant, average number of seeds per spike and number of seeds per spikelet, average number of spikes and average number of seeds per plant, average number of spikes and seed weight per plant, average spike length and average number of seeds per plant, average spike length and seed weight per spike, average spike length and seed weight per plant, average number of spikelets per spike and average number of seeds per plant, average number of spikelets per spike and seed weight per spike, average number of spikelets per spike and seed weight per plant, average number of seeds per spike and average number of seeds per plant, average number of seeds per spike and seed weight per spike, average number of seeds per spike and seed weight per plant, number of seeds per spikelet and average number of seeds per plant, number of seeds per spikelet and seed weight per spike, number of seeds per spikelet and seed weight per plant, average number of seeds per plant and seed weight per spike, average number of seeds per plant and seed weight per plant, seed weight per spike and seed weight per plant as well as seed weight per spike and thousand kernel weight (Tables S8–S10). The negative correlation coefficients were observed in all three years of study between the average number of spikelets per spike and pith in uppermost internode, plant height and average number of seeds per spike, average stem filling with pith in total internodes and average number of seeds per spike, plant height and number of seeds per spikelet, pith in uppermost internode and average number of seeds per plant, pith in uppermost internode and seed weight per spike, pith in uppermost internode and seed weight per plant, average stem filling with pith in total internodes and average number of seeds per plant, average stem filling with pith in total internodes and seed weight per spike as well as average stem filling with pith in total internodes and seed weight per plant (Tables S8–S10).

## 4. Discussion

The research results showed that the expression of the stem-solidness was not always consistent with the data provided in scientific publications, on the websites of gene banks and plant breeding companies. According to the reference publications, two Polish cultivars (Arabeska and Nawra) were described as hollowed stemmed. However, in our study they exhibited minimal expression of the trait (6.5 and 7.6 points, respectively). The Lilian cultivar, which is native to Canada, is considered one of the best solid-stemmed varieties [17,24]. However, our study showed that in Poland, it was characterised by intermediate stem filling with parenchyma (7.2 points).

The high diversity of genotypes and the weather conditions during the growing seasons significantly affected the expression of important agronomic traits, including the stem filling. This is consistent with the results obtained by De Puaw and Read [23], Weiss and Morill [15], Beres et al. [17,18], and Nilsen et al. [19] who found that the stem-solidness depends on rainfall, air temperature, the amount of light that reaches the plants, and plant density per m$^2$. The results of our study partially differed from the data describing the characteristics of the varieties, which were provided on the websites of the gene banks. Contrary to the information given on these websites, the stems of some cultivars were not filled with pith. The spring wheat genotypes in our collection were mathematically divided into seven groups differing in the degree of stem-solidness.

There were only nine cultivars with a very high degree of stem filling (AC Abbey, AC Eatonia, Canuck, Carola, Cltr 7027, Cltr 7028, MT 776, Rescue, and Tybalt). The Rescue and AC Abbey genotypes derive their solidness from the Portuguese landrace line, 'S-615' [25], and have an allele that results in stem durability [19].

In contrast to the data in reference publications, our experiments showed that some varieties did not develop pith in the Polish environmental condition. According to the information provided by the IPK Gatersleben gene bank, the stems of the German variety Heines Germania should be filled with parenchyma to a large extent. However, due to the different conditions of cultivation than in its place of origin and due to the influence of weather in individual years of research, the stems of this cultivar did not develop pith inside. In some cases, the weather conditions resulted in a lower expression of this trait. For example, the solid-stemmed Lillian cultivar, which is commonly grown in Canada due to its resistance to wheat stem sawflies, high yield and high protein content in seeds [17,24] did not exhibit a strong expression of this trait in Poland and was classified into the group with stems intermediately filled with pith. Subedi et al. [26] also noted the inconsistent pith expression in this cultivar.

The genotypes analysed in our study differed not only from each other but also within the same cultivar and exhibited variability in individual growing seasons. The influence of cultivation conditions and weather on stem-solidness has been confirmed many times. Various studies have proven that the expression of the trait depends on rainfall, air temperature, the amount of light that reaches the plants [1,15], and plant density [16,17]. In order to eliminate the trait variability under the influence the latter parameter, a density of 350 seeds per $m^2$ was used in our experiment. The choice of the value of this parameter was based on the results of a study by Beres et al. [17], who observed the most stable filling of the stem with pith at a density of 205–350 plants per $m^2$. This is due to the fact that plants shade one another [27], and the amount of light during the stem elongation phase significantly affects the formation of pith [28]. The analysis of the results showed that the degree of stem depended on the weather conditions in a particular growing season. This was largely due to the significant interaction of the genotype with the environment. Only 5 of the 50 cultivars under analysis exhibited stability of the trait regardless of the environmental conditions. These were two solid-stemmed genotypes (AC Eatonia, Rescue), two hollow-stemmed genotypes (Bombona and Łagwa), and one cultivar with intermediately filled stems (401). The stability of the AC Eatonia genotype was confirmed in a study by Subedi et al. [26].

The difficulties in interpreting the obtained results are related to the inheritance of the solid stem. There are several different proposals of inheriting stem solidness in the literature. In the 1960s, the possibility of inheritance from one major recessive gene and one or two minor genes, to four genes (one of which is epistatic to the others) was suggested [29,30]. Nowadays, molecular markers have been used to analyse the inheritance of this trait. Cook et al. [31] found a single solid stem QTL (*Qss.msub-3BL*) on chromosome 3BL that contributed at least 76% of the total variation for stem solidness in the analysed DH population derived from a 'Rampart' (solid stems) × 'Jerry' (hollow stems) cross. Lanning et al. [32] identified a locus on chromosome 3D (*Qss.msub-3DL*), that controlled 31% of the variation for solidness in the cross between a solid-stemmed line and a line with intermediate solidness. The strong expression of steam solidness in Choteau is related with presence *Qss.msub-3BL* and *Qss.msub-3DL* [32]. After analysing nine F2 populations (from the crosses solid stem × hollow stem and hollow stem × hollow stem genotypes), Bainsla et al. [33] found multiple complementary factors in the additive fashion for stem solidness.

Although the breeding of solid-stemmed varieties is the best form of protection against wheat stem sawfly, for a long time it was believed that it was a trait that negatively affects grain yield [20]. However, many contemporary studies emphasise the lack of negative correlations between stem solidness and yield [31,32]. Another noteworthy feature is the lodging problem. As modern wheat cultivars have a relatively ideal plant height [34], the reduction in the lodging index can be obtained by increasing the stem strength. Breeders

should pay attention to features such as thick stem wall, high lignin content, and stem diameter [35,36]. Our analysis of stem-solidness may help to make decisions about the choice of breeding materials. The research findings have already been used to select parental components for crossing cultivars characterised by a low variability of stem-solidness.

## 5. Conclusions

The UPOV methodology used for the analysis of the filling of the uppermost internode with pith gave the same results as the DePauw and Read methodology [23] used for the assessment of the stem filling in the whole plant based on the cross-section of the first (uppermost) internode. The assessment of the stem filling according to the DePauw and Read methodology [23] showed that the internodes in the lower part of the plants were filled with pith to the greatest extent. This methodology also shows a more precise distribution of the parenchyma tissue inside the entire stem than the UPOV methodology, which is used only for the assessment of the uppermost internode.

Having conducted the experiment in Poland, nine Polish cultivars, two Canadian genotypes (BW 597 and AC Elsa), and one Portuguese genotype (I 836) were classified as hollow-stemmed. Regardless of the year of the research, the Cltr 7027 genotype was filled with pith to the greatest extent. The analysis of most genotypes described in reference publications as solid-stemmed showed that when grown in Poland, their stems were only partly filled with pith. The differences in the stem-solidness in the same genotypes in individual years of the study show that this trait is influenced by the environmental conditions. Our research finding confirmed the observation made by Bainsla et al. [31], who noted that the filling of the stem with pith is a complex trait, with a high variability of expression.

**Supplementary Materials:** The following are available online at https://www.mdpi.com/article/10.3390/agronomy11081640/s1. Table S1. The collection of spring wheat genotypes (Agricultural Research Station Dłoń, 2012–2014), Table S2. Mean squares from combined analysis of variance in randomised complete block design of 50 spring wheat genotypes for 18 quantitative traits in 2012–2014, Table S3. The average stem filling with pith in individual internodes in the collection of 50 spring wheat genotypes (Experiment 1—Agricultural Research Station Dłoń, 2012–2014), according to the methodology used in the USA, Table S4. The influence of the weather conditions in consecutive research years on the stem filling with pith in 50 spring wheat genotypes (Agricultural Research Station Dłoń, 2012–2014), Table S5. The average plant height and the number of stems in 50 spring wheat genotypes (Agricultural Research Station Dłoń, 2012–2014), Table S6. The average number of spikes and spike length in 50 spring wheat genotypes (Agricultural Research Station Dłoń, 2012–2014), Table S7. The average seed weight per plant and the mean thousand kernel weight in 50 spring wheat genotypes (Agricultural Research Station Dłoń, 2012–2014), Table S8. The correlation coefficients between spring wheat genotypes traits observed in 2012, Table S9. The correlation coefficients between spring wheat genotypes traits observed in 2013, Table S10. The correlation coefficients between spring wheat genotypes traits observed in 2014.

**Author Contributions:** Conceptualisation, J.N.; data curation, M.P. and D.K.-P.; formal analysis, D.K.-P. and J.B.; investigation, M.P.; methodology, D.K.-P., J.N. and J.B.; resources, M.P., D.K.-P., and J.N.; software, M.P., D.K.-P., J.N. and J.B.; supervision, J.N.; visualisation, J.N. and J.B.; writing—original draft, M.P., D.K.-P. and S.M.; writing—review and editing, J.B. and S.M. All authors have read and agreed to the published version of the manuscript.

**Funding:** This publication is being financed by the framework of Ministry of Science and Higher Education program as "Regional Initiative Excellence" in years 2019–2022, project no. 005/RID/2018/19.

**Institutional Review Board Statement:** Not applicable.

**Informed Consent Statement:** Not applicable.

**Data Availability Statement:** Not applicable.

**Conflicts of Interest:** The authors declare no conflict of interest.

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
