# Peer review of "Estimation of Stem-Solidness and Yield Components in Selected Spring Wheat Genotypes"

_agronomy, doi:10.3390/agronomy11081640_

Round 1

Reviewer 1 Report

The manuscript (ms) assess the stem solidness of fifty spring wheat cultivars with pith repeated for three years in Poland. The topic of this work is very interesting and novel however, as it was pin pointed to the first review, the submitted manuscript should have been substantially revised before being accepted for publication The authors failed to address appropriately most of the comments and suggestions of the first review. The authors need to substantially revised their ms based on the comments of the 2nd review before been accepted for publication. 

Reviewer 2 Report

The study explores about the parenchima content in wheat stem in relation to genetic and environmental variability.  The study, conducted in Poland, deals with and international set of wheat genotypes. Different groups of genotypes have been individuated as the main outcome from this study. 
In general the manuscript is well written, however I think that more interesting information could be extracted.

First of all, N and P management could influence plant height, harvest index and biomass production. Please the authors detail agronomic information in terms of N fertilization and, possibly, timing. In addition, supplementary materials includes important data about yield and yield components, which could help to be put in relation with stem filling. I suggest to perform a screening correlation analysis with the observed parameters, in order to observe potential existing relationship within the different set of genotypes.

Also,  as stated by the authors, stem filling could positively influence crop defence from pests and lodging; was this assessed in the trials or no information is available? In case, please give some information about this.

Finally, in Figure 4 it is reported the range of variability of  stem-solidness: please detail in the Methods section how this was calculated (standard deviation / means = coefficient of variation?; else?). Some genotypes showed a variability due to environment, while others showed more stability. I think that this could be an interesting information, since this trait seems not completely regulated by genetic effect, as for how reported. The GxE interactions shall be highlighted. 

Then, relationships of stem filling with radiation, temperature and rainfall could be helpful (as linear/quadratic relationship - or also within a correlation matrix - even if this is a statistically different test). 

According to this, we suggest to a major revision, in order to improve the manuscript according to the indications suggested above

Reviewer 3 Report

The study conducted by Pluta et al provides a multi year assessment of stem-solidness trait in various local and exotic wheat germplasm. A brief connection of the trait with resistance to wheat stem sawfly was indicated in the introduction and previously developed methods were used to measure the trait. Results presented in this manuscript may be found suitable for publishing in Agronomy once after a careful English language revisions are made. No references are usually cited in the Abstract hence authors are advised to remove the cited literature from Abstract. Introduction section need to be thoroughly revised to make the background more clear. Various small sentences (Line 25-30) can be more clearly joined and rewritten to make a coherent paragraph. Following recommendations can help authors to improve the results presented in the study:

- Line 313: Please cite the 'reference publications' where these genotypes were tested and classified as hollowed-stemmed. 

  • It has been seen that the trait (stem-solidness) is influenced by environmental factors hence calculating Heritability for each individual year for all the genotypes can provide a better idea about the inheritance of the trait.
  • Authors have measured various other plant parameters from the genotypes and presented those in supplementary data. It would be helpful for readers to see how these traits are associated with the main tarit i.e stem-solidness. A correlation analysis would be more presentable in this case.

Round 2

Reviewer 2 Report

The authors carried out a revision according to the indications suggested. The outcomes from correlation analysis seem quite interesting. A negative correlation between stem filling and seed yield and yield components was clearly observed. I think that the energetic effort to increase parenchyma content in stem, on one hand improve resistance to lodging and insect attacks, on the other hand final production might be negatively influenced. 

Also, the results of the ANOVA reported in Table S2 suggest that the observed variability is mainly influenced by environment (crop year), rather than G and GxY. However, genotype influence is important. 

The comments into the manuscript about what stated above are sufficient.

As far as I am concerned, the manuscript resulted improved and acceptable to be published in peer review journal. 

Author Response

Author would like to thank reviewers for their essential comments and spending valuable time to review this manuscript.

This manuscript is a resubmission of an earlier submission. The following is a list of the peer review reports and author responses from that submission.

Round 1

Reviewer 1 Report

The manuscript deals with the topic of the steam filling for spring wheat genotypes. This is very interested but my version of submitted document is incomplete. Based on the text and supplementary file I made a review. I feel that the missed figures and tables make the text more clear. I think that the result section could be developed and the calculated LSD value should be fully used (detail comments, regarding line 229 comment). I also have some doubts about the meeting of MANOVA assumption (detail comments, regarding line 111 comment).

Because I don’t know what the missing figures and tables contain, my manuscript validation is major revision, though the adjustment may be easier to do.

Detail comments:

Lack of figures 1 to 4 and lack of tables 1 to 4. This lack made the review very hard. I suspect that after attaching such files the text will be more clear. I have moreover some comments:

Line 92: ‘30-50 plants’ why different number of plants was used?

Line 111: Which traits were tested for normal distribution? It was written that ‘the number of seeds per spikelet, the number of seeds per plant’ were denoted, but both cannot have an N distribution. The first one is the second one divided by the random number of spikelet per plant. My advice is to not use the redundant variables (indicators).

Line 113 and 114: what were the dependent variables and what were the independent variables for MANOVA and ANOVA?

Line 123: What is the value of ‘average 123 temperatures between 1961 and 2011’?

Line 126: What is the ‘average total rainfalls between 1961 and 2011’?

Lines 131 to 143: Are the names of cultivar in the table1? I have lack of tables. It is interesting which cultivars changed they classification.

Line 143: ‘were confirmed by a significant cultivar ×year interaction’ – from which analysis?

Line 146: I’m confused. ‘21 genotypes could be classified as intermediately filled’ – whereas (line 140) ‘12 were classified into the group with intermediate pith filling’ in the 2014. Maybe the table 1 contains the genotype –by- year classification? Based on the text I fill that the text is self-contradictory.

Line152: Line 79, the ‘Supplementary Table 1’ is cited, line 152 – ‘SM Table 6’. The supplementary tables 2 to 5 were not referred for this line. Moreover I don’t understand why this table is cited at this point.

Line 174: I don’t see the analysis table.

Line 178: It is lack of cluster analysis in the ‘methods’ section. Which traits were the base for this analysis? Were they transformed before (centered or standardized)? I have the lack of the table 2.

Line 187: ‘The information in the table shows very precisely …’ – I can’t review this part of the text because the lack of figures and tables.

Line 198: ‘SM Table 7’ – my supplementary material contains only 6 tables.

Line 206: ‘The results of biometric analyses can be found in the supplement to this study (SM Tables 2-4).’ – the tables contain the averaged observed values. After the words ‘results of analyses’ I suspected the ANOVA, MANOVA, Cluster Analysis or something similar. Maybe ‘The results of biometric measures’ will be better?

Line 241: ‘he spring wheat genotypes in our collection were mathematically divided into seven groups’ What means ‘mathematically divided’?

Line 229, whole ‘results’ section: Maybe the figures or tables contain more details about stem-solidness in wheat, but the text in results section should be developed in this point. I feel that the title has not been realized (despite the obtained results which seem to be sufficient). The rank of cultivars? The stability of cultivars? The LSD was not used. Is the cultivar A significantly more filled than cultivar B?

Moreover I feel unsatisfied due to the lack of information, what was the variability (standard deviation for example) for a given species under the same conditions (year) and what was caused by environmental factors. This is mentioned in the ‘discussion’ and ‘Conclusions’ section, but without calculations (maybe such results are in missed tables).

Line 275: ‘The UPOV methodology used for the analysis of the filling of the uppermost internode with pith gave the same results as the DePauw and Read methodology [1]’ and (line 279) ‘This methodology also shows a more precise distribution’ – I’m confused – the considered methods were in accordance or one was better?

Author Response

Dear Reviewer, in attached file is a replay to the comments to the manuscript.

Yours sincerely 

Sylwia Mikołajczyk

Reviewer 2 Report

The manuscript (ms) assess the stem solidness of fifty spring wheat cultivars with pith repeated for three years in Poland. The topic of this work is very interesting and novel. However, there are several issues the authors need to address. The authors need to properly analysis their data and describe properly all the statistical analysis carried out. The results should be discussed in the context of the existing literature and should not be concentrated only to the discrepancy between the results of the study and the passport data obtained from the gene banks. I strongly suggest to the authors to fully rewrite the ms. Therefore, a major revision is required for this work before it can be further considered for publication.

More specifically the authors should take into consideration and properly justify in the Statistical Analysis (section 2.4) the following points

  1. The data from the assessment of the stem solidness are qualitative (ordinal), therefore they are not continuous variables.
  2. The appropriate analysis of the experiments repeated in time is the Combined Analysis of Variance. This is the classical analysis carried out by the breeders in breeding experiments. Therefore, the two-way analysis of variance is not the appropriate method since two-way analysis requires repetition and randomization of the two factors.
  3. The description of the statistical analysis applied for the results presented at sections 3.3.1 and 3.3.2 are missing from the section 2.4.
  4. In several points and at the concussions (line 286) the authors stated that the environmental conditions can significantly affect stem solidness. However, the authors do not provide any statistical analysis to investigate the relations between stem solidness and environmental conditions. I highly recommend the authors to include such analysis.
  5. In several points and at the conclusions (Line 278) the authors stated that “the internodes in the lower part of the plants were filled with pith to the greatest extent”. However, the authors do not provide any statistical analysis to investigate if the differences between stem solidness at different internodes are significant. Therefore, their conclusion is weak. I highly recommend the authors to include such analysis.
  6. The authors measured several yield components. However, they did not provide any analysis to investigate the relation of yield components with stem solidness. Such analysis will add value to their work and I highly recommend the authors to elaborate on this point.
  7. The ANOVA tables should include the significance of the main effects and the interactions. It will also be interesting the authors to present the variance explained the main effects and the interactions for each trait.

Author Response

Dear Reviewer,

files with response to review, manuscript, supplement and additional tables are attached as pdf files.

Sincerely

correspondin author

Sylwia Mikołajczyk
